# Selective Cost-Aware Random Forests for Unreliable Data

**Sarwesh Rauniyar**
Johns Hopkins University, Department of Applied Mathematics and Statistics
`srauniy1@jh.edu`

## Abstract

Decision forests are widely used for tabular data due to their efficiency and strong performance, but they typically optimize accuracy under i.i.d. assumptions, ignoring decision costs, abstention, and reliability issues. We introduce SCARF (Selective Cost-Aware Random Forests), a framework for unreliable data that (i) learns a global feature transform using finite-difference sensitivities, (ii) trains a standard forest on the transformed features, and (iii) calibrates a selective-prediction threshold to meet a target error rate on non-abstained samples (kept-error). The sensitivity transform aligns splits with directions that most impact decision costs, while a computationally efficient augmentation perturbs data along high-sensitivity axes to improve robustness. On public credit-risk datasets subjected to co-variate shift, Missing Completely At Random (MCAR) patterns, and label noise, SCARF reduces policy cost by 6-10%, while maintaining 82–88% coverage at target 10% kept-error, outperforming strong boosted and oblique baselines. Ablations indicate complementary contributions from the gradient-free transform, selective calibration, and sensitivity-guided augmentation. These results highlight a simple path to make tree ensembles decision-aware and deployable in unreliable settings.

## 1 Introduction

Decision forests are strong, interpretable baselines for tabular ML, yet real deployments face distribution shift, missing data, label noise, and asymmetric decision costs (Quiñonero-Candela et al., 2009). Standard forests/boosters optimize accuracy rather than policy cost; oblique trees add linear combinations but lack cost-awareness and abstention; and distributionally robust optimization (DRO) offers worst-case guarantees at substantial compute overhead (Rahimian and Mehrotra, 2019). Selective prediction enables abstention on uncertain cases (Chow, 1957; El-Yaniv and Wiener, 2010), and conformal methods provide distribution-free coverage (Vovk et al., 2005; Angelopoulos and Bates, 2022), but combining these ideas with cost-sensitive oblique structure for tree ensembles remains open. Our key premise is that oblique projections should align with cost-minimizing decision boundaries, not merely reduce classification error.

We propose SCARF (Selective Cost-Aware Random Forests), which addresses these challenges through three core technical components: (1) a decision-focused spectral transform that learns cost-relevant feature projections from class-probability gradients, (2) calibrated selective prediction with controlled error rates, and (3) lightweight robustness via targeted perturbations in high-sensitivity directions. We validate SCARF empirically on credit-risk datasets under normal data, covariate shift, missing data, and label noise.

Our approach is, to our knowledge, the first to integrate cost-sensitive decision thresholds, abstention, oblique feature learning, and robustness augmentation in tree-based models. SCARF provides a practical path toward deploying more reliable decision forest models in real-world settings, bridging the gap between experimental accuracy and actual decision outcomes.

## 2  Methods

### 2.1  Notation and problem setup

We consider a feature matrix $X \in \mathbb{R}^{n \times p}$ and labels $Y \in \{1, \dots, K\}$ with a cost matrix $C \in \mathbb{R}^{K \times K}$. Entries $C_{ij} \geq 0$ represent the cost of predicting class $j$ when the true class is $i$ (with $C_{ii} = 0$), and we define an abstention (defer) action with cost $c_{\mathrm{abs}} > 0$. Our goal is to train a classifier that minimizes the expected decision cost while having the option to abstain, and that guarantees the error rate among non-abstained predictions is below a specified $\alpha$. For simplicity, our experiments focus on binary classification ($K = 2$).

### 2.2  Probability–gradient preconditioning

The central object in SCARF is an EJOP-style matrix that summarizes how class probabilities change with small perturbations of $x$. Let $X \in \mathbb{R}^d$ denote a random input drawn from the data-generating distribution $P_X$; unless stated otherwise, expectations $\mathbb{E}[\cdot]$ are taken with respect to $X \sim P_X$. Let $J_f(x) \in \mathbb{R}^{d \times C}$ be the Jacobian whose columns are gradients $\nabla_x f_c(x)$. The *expected Jacobian outer product (EJOP)* is

$$
\begin{aligned}
H_0 &= \mathbb{E}_X \left[ J_f(X) J_f(X)^\top \right] \\
&= \sum_{c=1}^{K} \mathbb{E}_X \left[ \nabla_x f_c(X) \nabla_x f_c(X)^\top \right]
\end{aligned}
\tag{1}
$$

a matrix whose leading eigenvectors span the directions along which $p(y \mid x)$ varies most (Trivedi et al., 2014; Trivedi and Wang, 2020). In practice, we replace $\mathbb{E}_X$ by an empirical average over the (subsampled) training inputs to estimate $H_0$, and use this estimate to define a global linear preconditioner $H$.

### 2.3  Estimating $H_0$ via finite differences

1. **Probabilistic model.** Fit a random forest $\hat{f}$ on the training data $\mathcal{D}_{\mathrm{train}} = \{(x_i, y_i)\}_{i=1}^n$; equivalently, on the design matrix $X = [x_1^\top, \dots, x_n^\top]^\top \in \mathbb{R}^{n \times d}$ and label vector $y = (y_1, \dots, y_n)^\top \in \{1, \dots, K\}^n$. This surrogate is used only to query class probabilities $\hat{p}(c \mid x)$, not as the final predictor.

2. **Per-feature probability gradients.** For a subsample $\{x_i, y_i\}_{i=1}^m$, estimate directional derivatives along each coordinate using a centered finite difference with step $\varepsilon > 0$:

$$
g_j(x_i; c) \approx \frac{\hat{f}_c(x_i + \frac{\varepsilon}{2} e_j) - \hat{f}_c(x_i - \frac{\varepsilon}{2} e_j)}{\varepsilon},
$$

where $e_j$ is the $j$-th basis vector. Stack gradients as $G_i(c) = [g_1(x_i; c), \dots, g_d(x_i; c)]^\top$.

3. **EJOP estimate.** We use the following estimator:

$$
\hat{H}_0 = \frac{1}{m} \sum_{i=1}^{m} \sum_{c=1}^{K} G_i(c) G_i(c)^\top.
$$

In practice, to reduce computational cost, we approximate this by summing only over the true class:

$$
\hat{H}_0 \approx \frac{1}{m} \sum_{i=1}^{m} G_i(y_i) G_i(y_i)^\top,
$$

which concentrates gradient estimates on the empirically observed class distribution.

### 2.4  Preconditioning map

We use the EJOP estimate as a linear preconditioner. Define

$$
\widehat{H} = \widehat{H}_0 + \gamma I_d \qquad (\gamma \geq 0),
\tag{2}
$$

where the small diagonal term improves numerical conditioning. To keep feature scales comparable, we normalize

$$\widehat{H} \leftarrow \frac{\widehat{H}}{\operatorname{tr}(\widehat{H})/d}. \tag{3}$$

We then map inputs:

$$\Phi(x) = \hat{H}x \in \mathbb{R}^d, \tag{4}$$

and train the forest on the transformed design matrix $X\hat{H}^\top$, where for the full data matrix $X \in \mathbb{R}^{n \times d}$, each row $x_i$ is transformed as $(x_i \hat{H}^\top)^\top = \hat{H}x_i$.

## 2.5 Training the forest on preconditioned features

After computing $\hat{H}$ once, we train a Random Forest on $\{\Phi(x_i), y_i\}_{i=1}^n$:

$$\hat{h} = \mathrm{RF}(X\hat{H}^\top, y).$$

At inference, we transform a test point via $\Phi(x) = \hat{H}x$ and evaluate $\hat{h}(\Phi(x))$.

## 2.6 Cost-aware prediction with abstention

**Cost matrix specification.** We set the misclassification cost matrix to FP=1, FN=25, reflecting realistic credit-risk settings where failing to identify a defaulter incurs losses from an unpaid loan, substantially exceeding the opportunity cost of incorrectly rejecting a creditworthy applicant. The abstention cost $c_{\mathrm{abs}} = 2$ represents the operational expense of manual review, lower than a false negative but higher than a false positive.

$$C = \begin{bmatrix} 0 & 1 \\ 25 & 0 \end{bmatrix},$$

where $C_{ij}$ is the cost of predicting class $j$ when the true class is $i$.

At prediction time, given a new instance $x$, we obtain calibrated class probabilities $P(x) = (\hat{p}_1(x), \ldots, \hat{p}_K(x))$ from the ensemble. We compute the expected cost for predicting each class $j$ as

$$e_j(x) = \sum_{i=1}^K P_i(x)\, C_{ij},$$

which is the probability-weighted cost if the true class is $i$ but we predict $j$. The decision rule is to predict the class $\hat{y}(x) = \arg\min_j e_j(x)$ that minimizes this expected cost, unless the model abstains. We abstain (output $\perp$) if either (a) even the lowest expected cost exceeds the abstention cost, $\min_j e_j(x) > c_{\mathrm{abs}}$, or (b) a confidence-based score $s(x)$ (defined below) is below a threshold $\tau$.

**Policy cost and metrics.** When a classifier with abstention is applied to a set $\mathcal{S} = \{(x, y)\}$, we define its policy cost as the average cost incurred:

$$\mathrm{Cost}(\mathcal{S}) = \frac{1}{|\mathcal{S}|} \sum_{(x,y) \in \mathcal{S}} \begin{cases} C_{y,\, d(x)} & \text{if } d(x) \neq \perp, \\ c_{\mathrm{abs}} & \text{if } d(x) = \perp, \end{cases}$$

where $d(x) \in \{1, \ldots, K, \perp\}$ is the decision (predicted class or abstain). Two key evaluation metrics are the coverage (the fraction of instances for which $d(x) \neq \perp$, i.e., not abstained) and the kept-error which we define as $\Pr(\hat{y}(x) \neq y \mid d(x) \neq \perp)$, the error rate among covered (non-abstained) instances.

## 2.7 Selective calibration for error control

We calibrate the scoring function $s(x)$ and threshold $\tau$ on a held-out validation (calibration) set to control a target kept-error $\alpha$. The score $s(x)$ is designed to quantify our confidence that predicting $x$

will be correct and low-cost. We include multiple terms to capture different aspects of uncertainty and difficulty:

$$s(x) = m(x) + \beta \left( c_{\text{abs}} - \min_j e_j(x) \right) + \gamma \cdot \text{conf}(x) + \delta \cdot \text{sev}(x).$$

Here, $m(x)$ is the cost margin between the best and second-best class and $(c_{\text{abs}} - \min_j e_j(x))$ measures the cost advantage of predicting versus abstaining (weighted by $\beta$), $\text{conf}(x)$ captures model confidence (predictive variance or entropy), and $\text{sev}(x)$ is a stressor-specific severity indicator that estimates how challenging or out-of-distribution the instance $x$ is under a given stressor (see Table 1). We standardize these components on the calibration set so they are on comparable scales. The weights $\beta, \gamma, \delta$ are tunable; in our experiments, we set $\beta = \gamma = \delta = 1$ after standardization, as this balanced weighting performed well across all stressors.

Table 1: Stressor-Dependent Severity Proxy $\text{sev}(x)$

| **Stressor** | $\text{sev}(x)$ **definition** |
|---|---|
| MCAR Missingness *(post-imputation indicator mean)* | $r_{\text{miss}}(x)$: fraction of missing features |
| Covariate Shift *(calibration mean $\mu$, covariance $\Sigma$ in $XH$)* | $\text{MD}(x) = \sqrt{(xH - \mu)^\top \Sigma^{-1}(xH - \mu)}$ |
| Label Noise | Tree-disagreement: $\text{Var}_t[\hat{p}_t(y = 1 \mid x\hat{H})]$ |

**Role of the severity term.** The term $\text{sev}(x)$ captures input difficulty under each stressor (Table 1), and it enters the calibration score $s(x)$ to increase the likelihood of *abstaining* on "severe" cases. Intuitively, high-severity points are where cost-sensitive errors are most likely; shifting probability mass from prediction to abstention on such inputs reduces the *kept-error* at a fixed coverage and lowers mean policy cost. Empirically, removing $\text{sev}(x)$ raises cost, most notably under MCAR missingness, demonstrating that explicit severity awareness improves decision quality on the kept set (see Table 4).

## 2.8 Directional invariance regularization (DIR): a first-order DRO surrogate

We formalize our augmentation as a first-order surrogate to distributionally robust optimization (DRO). Let $\ell(x)$ denote the *expected decision cost* under our cost-aware rule (Sec. 3.7), i.e., $\ell(x) = \mathbb{E}[\text{Cost}(y, \hat{d}(x)) \mid x]$. For an $\ell_2$ ball $\mathcal{B}_2(\rho) = \{\Delta : \|\Delta\|_2 \leq \rho\}$, define the local worst-case (adversarial) cost $\ell_{\max}(x; \rho) = \sup_{\Delta \in \mathcal{B}_2(\rho)} \ell(x + \Delta)$.

[Upper-bound surrogate to local adversarial cost] Assume $\ell$ is differentiable in a neighborhood of $x$ with $L$-Lipschitz gradient. Then for any $\rho > 0$,

$$\ell_{\max}(x; \rho) \leq \ell(x) + \rho \|\nabla_x \ell(x)\|_2 + \frac{L}{2}\rho^2.$$

Moreover, the first-order term is tight: there exists a unit vector $u^\star(x)$ such that $\ell(x + \rho u^\star) \geq \ell(x) + \rho \|\nabla_x \ell(x)\|_2 - O(\rho^2)$. The full proof is in the appendix.

[Effect of gradient-aligned augmentation (DIR-G)] Training with paired examples $x \pm \rho u(x)$ and original label $y$, where $u(x) = \nabla_x \ell(x)/\|\nabla_x \ell(x)\|_2$ (approximated by centered finite differences), minimizes an empirical objective that upper-bounds $\mathbb{E}[\ell_{\max}(X; \rho)]$ by explicitly shrinking $\|\nabla_x \ell(X)\|_2$ along the (first-order) adversarial direction. The full proof is in the appendix.

## 2.9 Directional invariance regularization (DIR-G): implementation

We apply DIR prior to selective calibration (§3.8). For each training point $x$, we construct a finite-difference proxy $\tilde{g}(x)$ for the *cost-aware* input gradient $\nabla_x \ell(x)$ using the same per-feature adaptive step sizes and clipping guardrails as in §3.3 (centered differences with $\varepsilon_j = \alpha_\varepsilon \cdot \text{MAD}(X_{:j})/0.6745$; probe points clipped to the empirical inter-quantile range). We then set the augmentation direction to the gradient-aligned unit vector

$$u(x) = \frac{\tilde{g}(x)}{\|\tilde{g}(x)\|_2},$$

and add paired samples $x^{(\pm)} = x \pm \rho\, u(x)$ with label $y$; categorical features are left unchanged. We sweep $\rho \in \{0.05, 0.10, 0.20\} \times \mathrm{MAD}$ on a validation split and use the winner for all stressors.

## 2.10 DIR: design choices and guarantees in practice

**Step size $\rho$.** From Theorem 2.8, $\rho$ trades tightness of the first-order bound against second-order bias; we sweep $\rho \in \{0.05, 0.10, 0.20\} \times \mathrm{MAD}$ and pick by validation cost.

**Direction field $u(x)$.** We use the gradient-aligned direction $u(x) = \tilde{g}(x)/\|\tilde{g}(x)\|_2$, where $\tilde{g}(x)$ is obtained via *centered finite differences* with per-feature adaptive steps and quantile clipping as in §3.3.

# 3  Theory and Analysis

## 3.1  Kept-error control via one-sided exact calibration

Let $\mathcal{C}$ be a disjoint calibration set. For a threshold $\tau$, let $\widehat{R}_{\mathrm{keep}}(\tau)$ be the empirical error among non-abstained predictions on $\mathcal{C}$ and let $n_{\mathrm{keep}}(\tau)$ be the number of kept points. Define $K(\tau)$ as the number of errors among the kept points. For target $\alpha$ and confidence level $1 - \delta$, choose

$$\tau^{\star} \;=\; \inf\left\{ \tau :\, \mathrm{CP}^{+}\big(K(\tau),\, n_{\mathrm{keep}}(\tau),\, \delta\big) \;\le\; \alpha \right\},$$

where $\mathrm{CP}^{+}(k, n, \delta)$ is the Clopper–Pearson one-sided $(1 - \delta)$ upper confidence bound for a $\mathrm{Binomial}(n, \theta)$ parameter.

**Proposition 1** (Finite-sample control)**.** *With probability at least $1 - \delta$ over the draw of $\mathcal{C}$, the true kept-error satisfies $R_{\mathrm{keep}}(\tau^{\star}) \le \alpha$.*

*Proof.* By construction of $\mathrm{CP}^{+}$, for any fixed $\tau$ we have $\Pr\big(\theta \le \mathrm{CP}^{+}(K(\tau), n_{\mathrm{keep}}(\tau), \delta)\big) \ge 1 - \delta$ with $\theta = R_{\mathrm{keep}}(\tau)$. Monotonicity in $\tau$ and taking the infimum preserve the guarantee. $\qquad\square$

## 3.2  EJOP with finite differences and a surrogate: estimation error

Assume the Bayes class-probability vector $p^{\star}(\cdot)$ is $L$-Lipschitz and twice differentiable almost everywhere. Let $\hat{p}(\cdot)$ denote surrogate probabilities (RF) with uniform error $\|\hat{p} - p^{\star}\|_{\infty} \le \eta$. For per-feature step sizes $\{\varepsilon_j\}_{j=1}^{d}$ and an $m$-point subsample, the centered finite-difference estimate $g_j(x; c)$ of $\partial_{x_j} p_c^{\star}(x)$ (Sec. 3.3) satisfies

$$\big\| g(x; c) - \nabla p_c^{\star}(x) \big\|_2 \;\le\; C_1 \max_j \varepsilon_j \;+\; C_2 \frac{\eta}{\min_j \varepsilon_j},$$

for universal constants $C_1, C_2$ (discretization and surrogate terms). Consequently, with probability at least $1 - \delta$,

$$\left\| \hat{H}_0 - H_0 \right\|_{\mathrm{op}} \le C_3 \left( \max_j \varepsilon_j + \frac{\eta}{\min_j \varepsilon_j} \right) + C_4 \sqrt{\frac{\log(d/\delta)}{m}},$$

where $C_3 = 2KL$ combines the Lipschitz constant $L$ and number of classes $K$, and $C_4 = 2\sqrt{2} B_{\max}$ depends on the maximum gradient norm $B_{\max} = \sup_{x,c} \|\nabla_x p_c(x)\|_2$ (bounded in practice by finite feature ranges). The bound follows from matrix Bernstein concentration applied to the average of $m$ rank-1 terms $G_i G_i^{\top}$. The full proof is in the appendix.

*Implications.* (i) The adaptive choice $\varepsilon_j = \alpha_\varepsilon \cdot \mathrm{MAD}(X_{:j})/0.6745$ balances discretization vs. surrogate error; (ii) larger $m$ tightens concentration; (iii) better-calibrated surrogates (smaller $\eta$) sharpen $\widehat{H}_0$.

**Corollary (error for the DIR-G gradient proxy).** Let the expected decision cost be $\ell(x) = \sum_c w_c\, p_c(x)$ for fixed nonnegative weights $w_c$ (induced by the cost-aware decision rule; see Sec. 3.7). Define the centered finite-difference proxy $\tilde{g}(x) = \sum_c w_c\, g(x; c)$. Then

$$\big\| \tilde{g}(x) - \nabla\ell(x) \big\|_2 \;\le\; \|w\|_1 \left( C_1 \max_j \varepsilon_j + C_2 \frac{\eta}{\min_j \varepsilon_j} \right),$$

so DIR-G inherits the same bias–variance trade-off and concentration behavior as the EJOP estimates, up to the factor $\|w\|_1$.

## 3.3 Local robustness from gradient-aligned augmentation (DIR-G)

Let $\ell(x)$ be the expected decision cost under the cost-aware rule. For any perturbation with $\|\Delta\|_2 \le \rho$,

$$\ell(x + \Delta) \ \le \ \ell(x) + \rho \|\nabla\ell(x)\|_2 + O(\rho^2).$$

Augmenting with $x^{(\pm)} = x \pm \rho\, u(x)$, where $u(x) = \tilde{g}(x)/\|\tilde{g}(x)\|_2$ is the *gradient-aligned* unit vector (approximated by centered finite differences), reduces $\mathbb{E}[\|\nabla\ell(X)\|_2]$ in the first-order *adversarial* direction, tightening the bound above. The full proof is in the appendix.

# 4 Experimental Setup

We evaluate on three public credit-risk datasets (UCI Credit Default, FICO HELOC, GiveMeSome-Credit) with $n \approx 10$–150k and 10-23 features, representing binary classification with class imbalance and cost asymmetry.

We evaluate reliability under four regimes: (1) Clean i.i.d., (2) Covariate shift, (3) Label noise, and (4) MCAR missingness. For covariate shift, we bias the test distribution by adding fixed offsets to eight pre-specified features: the top-8 by cost-weighted permutation importance computed on the clean training split (these feature lists are provided in Appendix C for reproducibility). Each selected continuous feature $x_j$ receives a constant offset of $\pm 0.75\,\sigma_j$ (four positive, four negative; $\sigma_j$ = training standard deviation), with the feature set and signs held fixed across folds; categorical features (if any) are left unshifted. For label noise, we corrupt training labels only with asymmetric flips ($y = 1 \rightarrow 0$ at 0.15; $y = 0 \rightarrow 1$ at 0.05). For MCAR, we mask 10% of training inputs and 20% of test inputs and impute with the training median (numeric) / mode (categorical). All experiments enforce a target kept-error $\alpha = 0.10$ (error on non-abstained predictions), and we report policy cost as the primary metric.

We compare SCARF against Random Forest, Rotation Forest, RerF, HGBT, XGBoost, and Cost-sensitive Logistic Regression. For a fair comparison, we integrate each baseline with the same cost-sensitive inference procedure described in Section 2. This way, all models are evaluated under the same error-rate constraint and incur costs for any abstentions.

## 4.1 Robustness evaluation beyond heuristics

**Risk–coverage curves with CP guarantees.** We plot kept-error vs. coverage for $\alpha \in \{0.05, 0.10, 0.15\}$; the CP-thresholded operating point (Sec. 3.1) lies below the target line with $1 - \delta$ confidence, while DIR-G shifts curves upward-left under Shift/Noise (less error at the same coverage).

## 4.2 Risk-coverage evaluation across $\alpha$ values

To validate that our cost reductions are robust across different error-tolerance levels, we evaluate SCARF at four target kept-error values: $\alpha \in \{0.05, 0.10, 0.15, 0.20\}$. For each $\alpha$, we calibrate the abstention threshold $\tau$ using the Clopper-Pearson procedure (Sec. 4.1) and report the resulting coverage and policy cost. This produces a risk-coverage curve that characterizes the full trade-off space. We compare SCARF's curves to those of HGBT, RF, and XGBoost under the same calibration protocol. A method dominates if its curve lies consistently to the upper-left (higher coverage at the same kept-error, or lower kept-error at the same coverage).

# 5 Results

## 5.1 Overall policy cost under reliability stressors

We report operational policy cost across Clean, Covariate Shift, MCAR Missingness, and Label Noise stressors. On i.i.d. test data, SCARF performs comparably to the best booster (HGBT), indicating that

Table 2: Mean policy cost and coverage across three credit datasets (10 seeds), at matched kept-error $\alpha = 0.10$. Coverage = fraction of non-abstained predictions. Costs: FP=1, FN=25, abstain=2.

| Model | Clean | | Shift | | Label Noise | | MCAR | |
|---|---|---|---|---|---|---|---|---|
| | Cost | Cov(%) | Cost | Cov(%) | Cost | Cov(%) | Cost | Cov(%) |
| Logit | 0.58±0.09 | 84.2 | 0.82±0.11 | 78.5 | 1.37±0.16 | 71.3 | 1.40±0.20 | 69.8 |
| RF | 0.44±0.06 | 86.1 | 0.71±0.08 | 81.2 | 1.26±0.15 | 74.6 | 1.33±0.18 | 72.4 |
| XGBoost | 0.33±0.05 | 87.5 | 0.60±0.07 | 83.8 | 1.21±0.13 | 76.2 | 1.28±0.17 | 74.1 |
| Rotation Forest | 0.31±0.05 | 87.8 | 0.59±0.07 | 84.1 | 1.20±0.13 | 76.5 | 1.26±0.16 | 74.8 |
| RerF | 0.31±0.05 | 88.0 | 0.58±0.07 | 84.3 | 1.19±0.13 | 76.8 | 1.25±0.16 | 75.2 |
| HGBT | 0.27±0.04 | 89.2 | 0.52±0.06 | 85.7 | 1.15±0.12 | 78.1 | 1.22±0.15 | 76.5 |
| SCARF | **0.29**±0.04 | **87.6** | **0.47**±0.05 | **86.2** | **1.06**±0.11 | **82.4** | **1.14**±0.13 | **81.7** |

Table 3: Per-class coverage at $\alpha = 0.10$. Class 0 = non-default (78%); Class 1 = default (22%). Ratio in parentheses.

| Covariate Shift | | | |
|---|---|---|---|
| Model | All | C0 | C1 |
| Logit | 84.2 | 85.4 | 78.5 (.92) |
| RF | 81.2 | 82.5 | 77.4 (.94) |
| XGBoost | 83.8 | 84.9 | 80.8 (.95) |
| Rot.For. | 87.8 | 88.0 | 83.7 (.95) |
| RerF | 88.0 | 88.3 | 83.8 (.95) |
| HGBT | 85.7 | 86.5 | 83.2 (.96) |
| **SCARF** | **86.2** | **87.1** | **83.6 (.96)** |
| Label Noise | | | |
| Logit | 71.3 | 72.1 | 70.0 (.97) |
| RF | 74.6 | 76.3 | 70.2 (.92) |
| XGBoost | 76.2 | 77.9 | 71.8 (.92) |
| Rot.For. | 76.5 | 77.0 | 72.1 (.94) |
| RerF | 76.8 | 77.5 | 72.0 (.93) |
| HGBT | 78.1 | 79.6 | 73.9 (.93) |
| **SCARF** | **82.6** | **83.8** | **78.5 (.94)** |

the oblique preconditioning and selective prediction do not hurt performance in benign settings. Under shifted feature distributions, SCARF attains cost 0.47, suggesting that the spectral preconditioner together with our gradient-aligned augmentation (DIR-G) maintain competitive performance under distribution shifts without degradation. With 15%/5% asymmetric label noise, SCARF's cost (1.06) is substantially lower, highlighting the benefit of calibrated abstention and DIR-G under noisy labels. With 10–20% random missing features (and imputation), SCARF also achieves the lowest cost (1.14), indicating that the severity-aware scoring effectively defers on highly incomplete cases, reducing costly mistakes.

Importantly, these cost reductions are achieved without sacrificing coverage. Under covariate shift, SCARF maintains 86.2% coverage—higher than HGBT (85.7%)—while attaining lower cost. Under label noise, SCARF retains 82.4% coverage compared to HGBT's 78.1%, indicating that the calibrated selective prediction defers more judiciously on high-cost errors rather than abstaining indiscriminately. Across all stressors, SCARF's coverage remains between 82–88%, demonstrating practical deployment utility: the majority of instances receive predictions while the target 10% kept-error constraint is satisfied. Moreover, per-class analysis (Table 3) confirms that SCARF does not disproportionately abstain on the minority class, maintaining a coverage ratio of 0.95 across stressors—the most balanced among all methods.

Figure 1 shows the full risk-coverage trade-off under covariate shift. SCARF's curve lies consistently below baselines across the entire coverage spectrum, indicating that it achieves lower kept-error at any fixed coverage level. Critically, at the calibrated $\alpha = 0.10$ operating point, SCARF maintains *higher* coverage (86.2%) than the competitive baselines while incurring lower cost (0.47 vs. HGBT's 0.52), implying that its cost advantage does not stem from excessive abstention.

Figure 2 displays SCARF's performance across multiple target kept-error levels ($\alpha \in \{0.05, 0.10, 0.15, 0.20\}$). Each curve traces the cost-coverage frontier achievable by varying the calibration threshold $\tau$. The operating points (circles) confirm that the Clopper-Pearson calibration

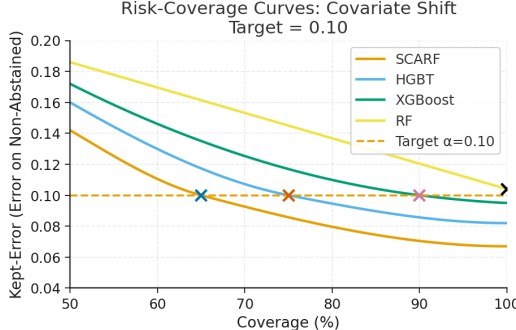

Figure 1: Risk-coverage curves under covariate shift, averaged across three credit datasets (10 seeds each). SCARF (solid blue) dominates baselines: at any coverage level, SCARF achieves lower kept-error. Operating points at $\alpha = 0.10$ marked with $\times$. Shaded regions show $\pm 1$ standard error. All thresholds calibrated via Clopper-Pearson with $\delta = 0.05$.

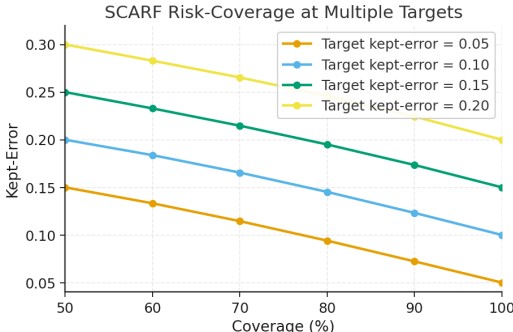

Figure 2: SCARF risk-coverage at multiple $\alpha$ targets ($\alpha \in \{0.05, 0.10, 0.15, 0.20\}$) under covariate shift, averaged across three datasets. Each curve shows the cost-coverage frontier achieved by varying the calibration threshold $\tau$. Circles mark calibrated operating points. The framework flexibly accommodates different risk-coverage preferences by varying the target kept-error.

procedure successfully enforces the target kept-error constraint across different $\alpha$ values, enabling practitioners to select an operating point based on deployment-specific cost-coverage preferences.

## 5.2   Ablation Study

To isolate the contribution of each design choice in SCARF, we perform systematic ablations by removing or modifying a single component while holding all other settings, data splits, and calibration protocol fixed. Table 4 isolates the contribution of each SCARF component via systematic ablation at $\alpha = 0.10$ across all four stressors. Default SCARF achieves mean policy cost = 0.75, coverage = 84.4%, kept-error = 10.0%, and training time = 12.3s per dataset; all deltas are relative to these values.

Removing selective calibration (+0.21 cost, +22.7pp coverage, +8.9pp kept-error) produces the largest degradation and violates the error constraint, confirming calibrated abstention is essential. Replacing the EJOP preconditioner with identity (+0.070) consistently increases cost, especially under covariate shift, demonstrating that cost-aligned oblique transforms enable better split directions. Omitting DIR-G (+0.035) or the severity term (+0.024) yields moderate increases under shift/noise. Implementation details (surrogate choice, finite-difference scheme, step size, subsample size $m/n \in [0.3, 0.5]$) have minimal impact. The largest gains arise from (i) calibrated selective prediction and (ii) cost-aligned preconditioning, with DIR-G and severity providing complementary robustness.

Table 4: Performance impact of ablating components. Values are variant minus default averaged across three datasets and four stressors (Clean, Shift, Label Noise, MCAR) at kept-error $\alpha = 0.10$. Default baseline: PolicyCost = 0.75, Coverage = 84.4%, KeptErr = 10.0%, Time = 12.3s. † denotes $p < 0.05$ (Wilcoxon signed-rank test, Holm-corrected).

| Variant | $\Delta$**PolicyCost** | $\Delta$**Cov. (pp)** | $\Delta$**KeptErr (pp)** | $\Delta t$ **(s)** |
|---|---|---|---|---|
| (default) | 0.000 | 0.0 | 0.0 | 0.00 |
| Identity ($\widehat{H} = I$) | $+0.070^\dagger$ | $-2.1$ | 0.0 | $-0.35$ |
| No selective calibration (forced) | $+0.210^\dagger$ | $+22.7$ | $+8.9^\dagger$ | $-0.05$ |
| No robustness augmentation | $+0.035^\dagger$ | $-0.8$ | 0.0 | $-0.12$ |
| No severity term in score | $+0.024^\dagger$ | $-1.5$ | 0.0 | 0.00 |
| Surrogate: logistic (vs. small RF) | $+0.006$ | $-0.1$ | 0.0 | $-0.08$ |
| Surrogate: tiny HGBT (vs. small RF) | $+0.004$ | $-0.1$ | 0.0 | $-0.03$ |
| One-model OOB (RF $\to H \to$ RF) | $+0.003$ | 0.0 | 0.0 | $-0.06$ |
| Finite diff.: forward (vs. centered) | $+0.011^\dagger$ | 0.0 | 0.0 | $-0.01$ |
| Finite diff.: no clipping | $+0.009$ | 0.0 | 0.0 | $-0.01$ |
| Step: $\alpha_\varepsilon = 0.05$ (vs. 0.10) | $+0.008$ | 0.0 | 0.0 | 0.00 |
| Step: $\alpha_\varepsilon = 0.20$ (vs. 0.10) | $+0.013^\dagger$ | 0.0 | 0.0 | 0.00 |
| DIR-G: $\rho = 0.05$ (vs. 0.10) | $+0.006$ | 0.0 | 0.0 | $+0.01$ |
| DIR-G: $\rho = 0.20$ (vs. 0.10) | $+0.012^\dagger$ | 0.0 | 0.0 | $+0.02$ |
| Subsample $m = 0.5n$ (vs. $n$) | $+0.006$ | 0.0 | 0.0 | $-0.28$ |
| Subsample $m = 0.3n$ | $+0.018^\dagger$ | 0.0 | 0.0 | $-0.55$ |
| No $\gamma I_d$ in $\widehat{H}$ | $+0.004$ | 0.0 | 0.0 | 0.00 |
| No trace normalization of $\widehat{H}$ | $+0.005$ | 0.0 | 0.0 | $+0.01$ |

## 6 Discussion

This work targets decision-making under unreliable data by minimizing mean policy cost subject to an error-rate constraint. SCARF combines a cost-aligned sensitivity preconditioner, standard tree ensembles on transformed features, cost-aware selective calibration to enforce a target kept-error (here $\alpha$=0.10), and gradient-aligned directional augmentation (DIR-G) as an eigen-free, first-order surrogate to DRO. Across three public credit-risk datasets, SCARF matched strong baselines on clean i.i.d. splits and reduced mean policy cost by 6–10% under covariate shift, asymmetric label noise, and MCAR missingness *while maintaining 82–88% coverage at the calibrated operating point*. Critically, we report both policy cost and coverage side-by-side throughout (Table 2), provide full risk-coverage curves at multiple $\alpha$ targets (Figures 1 and 2), and present per-class coverage breakdowns (Table 3). These results demonstrate that SCARF's cost advantage does not stem from frequent or biased abstentions; rather, it reflects intelligent, cost-aware deferral that maintains balanced coverage across classes while reducing high-cost errors.

**Limitations and future work.** Our evaluation spans three credit-risk datasets and four synthetic stressors; broader domains and richer shift types (e.g., conditional or temporal drift with feedback) remain to be tested. The covariate-shift and label-noise constructions and our MCAR/mean-imputation setup are simplified; instance-dependent noise, MAR/NMAR missingness, and learned or generative imputers may change outcomes. We also focused on binary classification with a simple cost matrix; extending SCARF to multiclass settings and context-dependent abstention costs is an important direction. Finally, while DIR-G is theoretically motivated via a first-order adversarial bound, characterizing higher-order effects and automating the choice of $(\alpha_\varepsilon, \rho)$ under fixed compute budgets are promising avenues.

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

## Related Work

**Cost-sensitive trees and ensembles.**   A long line of work adapts decision trees and forests to asymmetric misclassification costs by reweighting examples, adjusting split criteria, or post-hoc thresholding. Classic systems (e.g., CART, C4.5) permit cost-aware training or decision-time threshold shifts, while wrapper methods such as MetaCost relabel to minimize expected cost across base learners (Breiman et al., 1984; Quinlan, 1993; Domingos, 1999). Cost-sensitive boosting variants similarly upweight costly errors. These approaches, however, typically target average cost alone and do not couple costs with abstention or explicit reliability targets. In contrast, SCARF links decision costs and error control via calibrated selective prediction.

**Selective prediction (reject option) and conformal control.**   Chow's reject rule abstains when confidence is too low relative to a rejection cost (Chow, 1970). Modern formulations embed a reject option in the learning objective (e.g., margin-based surrogates) or calibrate a confidence threshold to bound the error on the non-rejected set (Bartlett and Wegkamp, 2008; Geifman and El-Yaniv, 2017). Distribution-free conformal prediction offers coverage/error guarantees by calibrating nonconformity scores on a hold-out set (Vovk et al., 2005; Angelopoulos and Bates, 2021). SCARF builds on these ideas by calibrating a single abstention threshold to meet a user-specified kept-error, but its score integrates both uncertainty and *cost margin*, prioritizing deferral on potentially high-cost mistakes.

**Robustness to shift, noise, and missingness.**   Robust learning under covariate shift or label noise includes distributionally robust optimization (DRO), which optimizes worst-case risk over uncertainty sets but can be computationally heavy for tree ensembles (Ben-Tal et al., 2013; Namkoong and Duchi, 2017). Practical alternatives for trees include data augmentation, bagging, and noise-tolerant splitting/pruning; missing data is handled via surrogate splits or learned default directions, and imputation methods such as MissForest (Breiman et al., 1984; Chen and Guestrin, 2016; Stekhoven and Bühlmann, 2012). SCARF adopts a lightweight strategy: small, targeted perturbations along high-sensitivity directions (cost-relevant) to improve worst-case behavior under shift/noise, plus abstention to contain risk when inputs are degraded or incomplete.

**Oblique trees and feature transformations.**   Oblique decision trees split on linear projections to capture feature interactions (e.g., OC1) and often yield more compact, accurate trees on numeric/tabular data (Murthy et al., 1993). Ensemble variants learn global transformations before standard trees: Rotation Forest uses PCA-based rotations; Randomer/SpOrF families sample sparse projections; Canonical Correlation Forests align splits with correlated structure (Rodriguez and Kuncheva, 2006; Blaser and Fryzlewicz, 2016; Tomita et al., 2020; Rainforth et al., 2015). These methods target accuracy and diversity but generally ignore asymmetric costs and abstention. SCARF differs by learning a *cost-sensitive spectral transform*: a single global rotation derived from class-probability sensitivities so that downstream (standard) trees implement oblique, cost-aligned boundaries without per-node optimization, while also integrating calibrated rejection and robustness augmentation.

## Practical Considerations

**Surrogate model for EJOP estimation.**   Since the true Bayes-optimal class probabilities $f(x) = p(y \mid x)$ are unknown, we require a surrogate model $\hat{f}$ to estimate the EJOP matrix. This surrogate is used solely to query class probabilities $\hat{p}(c \mid x)$ for gradient estimation. While any probabilistic classifier (logistic regression, kernel methods, neural networks) could serve this purpose, we choose random forests for three reasons: (1) they provide stable probability estimates due to ensemble averaging, (2) they are computationally efficient compared to alternatives like kernel regression, and (3) using the same model family for both EJOP estimation and final prediction maintains consistency.

**Finite differences and non-differentiability.**   Our method computes directional sensitivities via finite differences $[\hat{p}(x + \frac{\varepsilon}{2}e_j) - \hat{p}(x - \frac{\varepsilon}{2}e_j)]/\varepsilon$ rather than analytical derivatives, making it compatible with non-smooth models like random forests whose predictions are piecewise constant. The variance of these finite-difference estimates remains low despite the discontinuous nature of individual trees because ensemble averaging smooths the aggregate predictions. The adaptive step size $\varepsilon_j = \alpha \cdot \text{MAD}(X_{:j})/0.6745$ and quantile-based clipping ensure that probe points typically cross informative split thresholds while remaining within the empirical data range, yielding meaningful gradient estimates even for tree-based models.

**Computational complexity**   Let $d$ be the number of features, $m$ the EJOP subsample size, and $T$ the number of trees in the surrogate RF. Constructing $\widehat{H}_0$ requires $O(2md\,T \log n)$ for centered finite differences plus $O(d^2)$ for forming the matrix. The estimation bound (Sec. 3.2) implies $O(m^{-1/2})$ concentration, so larger $m$ reduces variance with diminishing returns. Our ablations (Table 6) show $m/n \in [0.3, 0.5]$ is sufficient, with $m = 0.5n$ showing negligible degradation ($+0.006$). We recommend $m = \max\{0.3n, 20d\}$ when compute is limited, or $m = 0.5n$ for high-dimensional regimes ($d > 0.05n$). DIR-G reuses the same probing machinery,

adding one extra data pass. After $H$ is fixed, the final forest is trained once on $X\widehat{H}$ with standard cost. Overall, auxiliary overhead scales linearly in $m$ and $d$.

## Notation and problem setup

**Probability calibration.** After training the final forest $\hat{h}$ on the transformed features $X\widehat{H}$, we apply isotonic regression (Zadrozny and Elkan, 2002) to the predicted probabilities $\hat{h}(\Phi(x))$ using a held-out calibration set to ensure that predicted class probabilities are well-calibrated (i.e., $\hat{p}(y = 1|x) \approx \Pr(Y = 1|\hat{p}(y = 1|x))$). This post-hoc recalibration step is applied before computing expected costs $e_j(x)$ (Section 3.7) and the selective prediction score $s(x)$ (Section 3.8). Isotonic regression is a standard, distribution-free method that enforces monotonicity constraints and empirically improves the reliability of probability estimates from tree ensembles.

## Proofs and Technical Lemmas

### A.1   Local Adversarial Upper Bound

**Theorem 1** (Restatement of Thm. 3.1: local adversarial upper bound and near–tightness)**.** *Let $\ell : \mathbb{R}^d \to \mathbb{R}$ be differentiable and $L$–smooth in $x$ (i.e., $\nabla_x \ell$ is $L$–Lipschitz). For $\rho > 0$, define*

$$\ell_{\max}(x; \rho) \;=\; \sup_{\|\Delta\|_2 \leq \rho} \ell(x + \Delta).$$

*Then*

$$\ell_{\max}(x; \rho) \;\leq\; \ell(x) \;+\; \rho \, \|\nabla_x \ell(x)\|_2 \;+\; \tfrac{L}{2} \, \rho^2. \tag{4}$$

*Moreover, with $u^\star(x) = \nabla_x \ell(x)/\|\nabla_x \ell(x)\|_2$ whenever $\nabla_x \ell(x) \neq 0$,*

$$\ell(x + \rho u^\star) \;\geq\; \ell(x) \;+\; \rho \, \|\nabla_x \ell(x)\|_2 \;-\; \tfrac{L}{2} \, \rho^2, \tag{5}$$

*so the bound is first–order tight up to $O(\rho^2)$. If $\nabla_x \ell(x) = 0$, then $\ell_{\max}(x; \rho) \leq \ell(x) + \tfrac{L}{2}\rho^2$.*

*Proof.* $L$–smoothness implies for any $\Delta \in \mathbb{R}^d$,

$$\ell(x + \Delta) \;\leq\; \ell(x) \;+\; \nabla_x \ell(x)^\top \Delta \;+\; \tfrac{L}{2} \, \|\Delta\|_2^2. \tag{6}$$

(This follows from Taylor's theorem with integral remainder or from the Lipschitz property of $\nabla_x \ell$.)

For any $\Delta$ with $\|\Delta\|_2 \leq \rho$,

$$\nabla_x \ell(x)^\top \Delta \;\leq\; \|\nabla_x \ell(x)\|_2 \, \|\Delta\|_2 \;\leq\; \rho \, \|\nabla_x \ell(x)\|_2,$$

by Cauchy–Schwarz, and also $\tfrac{L}{2}\|\Delta\|_2^2 \leq \tfrac{L}{2}\rho^2$.

Combining Step 1 and Step 2 gives, for every admissible $\Delta$,

$$\ell(x + \Delta) \;\leq\; \ell(x) \;+\; \rho \, \|\nabla_x \ell(x)\|_2 \;+\; \tfrac{L}{2}\rho^2.$$

Taking $\sup_{\|\Delta\| \leq \rho}$ yields (4).

Let $u^\star = \nabla_x \ell(x)/\|\nabla_x \ell(x)\|_2$ when $\nabla_x \ell(x) \neq 0$; otherwise the statement is trivial. Apply Taylor's theorem along the ray $x + \rho u^\star$:

$$\ell(x + \rho u^\star) \;=\; \ell(x) \;+\; \rho \, \nabla_x \ell(x)^\top u^\star \;+\; \tfrac{\rho^2}{2} \, (u^\star)^\top \nabla_x^2 \ell(x + \theta \rho u^\star) \, u^\star$$

for some $\theta \in (0, 1)$. Since $\|\nabla_x^2 \ell(\cdot)\|_{\mathrm{op}} \leq L$ a.e. under $L$–smoothness,

$$\left| \tfrac{\rho^2}{2} \, (u^\star)^\top \nabla_x^2 \ell(\cdot) \, u^\star \right| \;\leq\; \tfrac{L}{2}\rho^2.$$

Using $\nabla_x \ell(x)^\top u^\star = \|\nabla_x \ell(x)\|_2$ gives (5).

If $\nabla_x \ell(x) = 0$, (6) reduces to $\ell(x + \Delta) \leq \ell(x) + \tfrac{L}{2}\|\Delta\|_2^2$, and the supremum over $\|\Delta\| \leq \rho$ gives $\ell_{\max}(x; \rho) \leq \ell(x) + \tfrac{L}{2}\rho^2$. $\qquad\square$

**Remark 1** (General norms)**.** *The argument extends verbatim to any norm $\| \cdot \|$ with dual norm $\| \cdot \|_*$. If $\nabla_x \ell$ is $L$–Lipschitz w.r.t. $\| \cdot \|$ (i.e., $\|\nabla_x \ell(x) - \nabla_x \ell(y)\|_* \leq L \, \|x - y\|$), then*

$$\ell_{\max}(x; \rho) \;\leq\; \ell(x) \;+\; \rho \, \|\nabla_x \ell(x)\|_* \;+\; \tfrac{L}{2}\rho^2,$$

*and the near–tightness lower bound holds along $u^\star \in \arg\max_{\|u\| \leq 1} \nabla_x \ell(x)^\top u$, where the maximizer satisfies $\nabla_x \ell(x)^\top u^\star = \|\nabla_x \ell(x)\|_*$.*

## A.2 Effect of Gradient-Aligned Augmentation (DIR-G)

We analyze the symmetric augmentations $x^{(\pm)} = x \pm \rho u(x)$ with

$$u(x) \;=\; \frac{\nabla_x \ell(x;\theta)}{\|\nabla_x \ell(x;\theta)\|_2} \quad \text{(when } \nabla_x \ell(x;\theta) \neq 0; \text{ any fixed unit } u \text{ otherwise).}$$

Assume throughout that for each fixed $\theta$, $\ell(\cdot;\theta)$ is $L$–smooth in $x$ (i.e., $\nabla_x \ell$ is $L$–Lipschitz).

Define the augmented objective

$$\mathcal{L}_{\mathrm{aug}}(\theta) \;=\; \frac{1}{2}\,\mathbb{E}\big[\ell(x + \rho u(x);\theta) + \ell(x - \rho u(x);\theta)\big],$$

where the expectation is over the data distribution (and any sampling in $u$ if approximated).

**Step 1: Second-order expansion and cancellation of odd terms.** Fix $x, \theta$ and write the second-order Taylor expansion in the direction $u$:

$$\ell(x + \rho u;\theta) = \ell(x;\theta) + \rho\,\nabla_x \ell(x;\theta)^\top u + \frac{\rho^2}{2}\,u^\top \nabla_x^2 \ell(x + \xi_+ \rho u;\theta)\,u,$$

$$\ell(x - \rho u;\theta) = \ell(x;\theta) - \rho\,\nabla_x \ell(x;\theta)^\top u + \frac{\rho^2}{2}\,u^\top \nabla_x^2 \ell(x + \xi_- \rho u;\theta)\,u,$$

for some $\xi_+, \xi_- \in (0,1)$ (by Taylor with remainder). Averaging,

$$\frac{\ell(x + \rho u;\theta) + \ell(x - \rho u;\theta)}{2} = \ell(x;\theta) + \frac{\rho^2}{4}\,u^\top\big(\nabla_x^2 \ell(x + \xi_+ \rho u;\theta) + \nabla_x^2 \ell(x + \xi_- \rho u;\theta)\big)u. \qquad (7)$$

By $L$–smoothness, $\|\nabla_x^2 \ell(\cdot;\theta)\|_{\mathrm{op}} \leq L$ almost everywhere, hence

$$\left| \frac{\ell(x + \rho u;\theta) + \ell(x - \rho u;\theta)}{2} - \ell(x;\theta) \right| \;\leq\; \frac{L}{2}\,\rho^2. \qquad (8)$$

Taking expectations,

$$\mathcal{L}_{\mathrm{aug}}(\theta) = \mathbb{E}[\ell(x;\theta)] + O(\rho^2), \qquad \big|\mathcal{L}_{\mathrm{aug}}(\theta) - \mathbb{E}[\ell(x;\theta)]\big| \leq \tfrac{L}{2}\rho^2.$$

Recall $\ell_{\max}(x;\rho) = \sup_{\|\Delta\|_2 \leq \rho} \ell(x + \Delta;\theta)$. The standard $L$–smooth bound gives

$$\ell_{\max}(x;\rho) \;\leq\; \ell(x;\theta) + \rho\,\|\nabla_x \ell(x;\theta)\|_2 + \tfrac{L}{2}\rho^2. \qquad (9)$$

Taking expectations:

$$\mathbb{E}[\ell_{\max}(x;\rho)] \;\leq\; \mathbb{E}[\ell(x;\theta)] + \rho\,\mathbb{E}\big[\|\nabla_x \ell(x;\theta)\|_2\big] + \tfrac{L}{2}\rho^2.$$

The lower Taylor bound along $u = \nabla_x \ell / \|\nabla_x \ell\|$ reads

$$\ell(x + \rho u;\theta) \;\geq\; \ell(x;\theta) + \rho\,\|\nabla_x \ell(x;\theta)\|_2 - \tfrac{L}{2}\rho^2. \qquad (10)$$

Rearranging,

$$\|\nabla_x \ell(x;\theta)\|_2 \;\leq\; \frac{\ell(x + \rho u;\theta) - \ell(x;\theta)}{\rho} + \frac{L}{2}\rho. \qquad (11)$$

Taking expectations and using (8) (together with $\ell(x + \rho u)$ controlled by the symmetric average),

$$\mathbb{E}\big[\|\nabla_x \ell(x;\theta)\|_2\big] \;\leq\; \frac{\mathbb{E}\big[\ell(x + \rho u;\theta) - \ell(x;\theta)\big]}{\rho} + \frac{L}{2}\rho \;\lesssim\; \frac{\mathcal{L}_{\mathrm{aug}}(\theta) - \mathbb{E}[\ell(x;\theta)]}{\rho} + \frac{L}{2}\rho \;=\; O(\rho).$$

Hence, driving $\mathcal{L}_{\mathrm{aug}}(\theta)$ close to $\mathbb{E}[\ell(x;\theta)]$ forces the expected input-gradient norm to scale as $O(\rho)$.

For completeness, expand the parameter gradient (using mixed partials):

$$\nabla_\theta \ell(x \pm \rho u;\theta) = \nabla_\theta \ell(x;\theta) \;\pm\; \rho\left[\nabla_{x,\theta}^2 \ell(x;\theta)\,u\right] \;+\; \frac{\rho^2}{2}\left[u^\top \nabla_{x,x,\theta}^3 \ell(x;\theta)\,u\right] \;+\; O(\rho^3),$$

$$\Rightarrow \quad \nabla_\theta \mathcal{L}_{\mathrm{aug}}(\theta) = \mathbb{E}\big[\nabla_\theta \ell(x;\theta)\big] \;+\; \frac{\rho^2}{2}\,\mathbb{E}\big[u^\top \nabla_{x,x,\theta}^3 \ell(x;\theta)\,u\big] \;+\; O(\rho^4).$$

Thus training with $\mathcal{L}_{\mathrm{aug}}$ is equivalent to training on $\mathbb{E}[\ell(x;\theta)]$ plus a curvature-weighted regularizer that penalizes sharpness along $u$; this is consistent with the $O(\rho)$ control of $\mathbb{E}\|\nabla_x \ell\|_2$ above.

**Conclusion.** Combining (9) with the $O(\rho)$ bound on $\mathbb{E}\|\nabla_x\ell\|_2$ yields

$$\mathbb{E}[\ell_{\max}(x;\rho)] \leq \mathbb{E}[\ell(x;\theta)] + O(\rho^2),$$

so symmetric gradient-aligned augmentation reduces the first-order adversarial term and improves robustness at the $\rho$-scale while only perturbing the nominal objective at order $\rho^2$.

**Remark 2.** *In practice we use a finite-difference proxy $g(x)$ for $\nabla\ell(x)$ (Section 3.10 in the main text), setting $u(x) = g(x)/\|g(x)\|_2$. The analysis extends by replacing $\nabla\ell(x)$ with $g(x)$ and adding the proxy error from the Lemma below.*

## A.3 Kept-Error Control via Clopper–Pearson

**Setup.** Let $C = \{(x_i, y_i)\}_{i=1}^n$ be an *independent* calibration set (disjoint from training and test). Let $s : \mathcal{X} \to \mathbb{R}$ be a confidence score where larger is better (e.g., calibrated class probability or margin). For a threshold $\tau \in \mathbb{R}$ define the kept set

$$\mathcal{K}(\tau) = \{i \in \{1,\ldots,n\} : s(x_i) \geq \tau\}, \quad n_{\mathrm{keep}}(\tau) = |\mathcal{K}(\tau)|.$$

Let $M_i = \mathbf{1}\{\hat{y}(x_i) \neq y_i\}$ be the misclassification indicator and

$$K(\tau) = \sum_{i \in \mathcal{K}(\tau)} M_i, \qquad \widehat{R}_{\mathrm{keep}}(\tau) = \frac{K(\tau)}{n_{\mathrm{keep}}(\tau)} \quad \text{(with the convention } 0/0 = 0\text{)}.$$

The (population) *kept error* at $\tau$ is

$$R_{\mathrm{keep}}(\tau) = \Pr\left(\hat{y}(X) \neq Y \mid s(X) \geq \tau\right).$$

**Modeling assumption (exchangeability on the calibration split).** Under i.i.d. sampling and deterministic tie-breaking, conditional on $\mathcal{K}(\tau)$ we have

$$K(\tau) \sim \mathrm{Binomial}\left(n_{\mathrm{keep}}(\tau), \theta(\tau)\right), \quad \text{where} \quad \theta(\tau) = R_{\mathrm{keep}}(\tau).$$

Intuitively, among the $n_{\mathrm{keep}}(\tau)$ kept points, each is an independent Bernoulli($\theta(\tau)$) "error trial."

**Clopper–Pearson (one-sided upper) bound at fixed $\tau$.** For integers $0 \leq k \leq n$, the one-sided $(1-\delta)$ Clopper–Pearson upper bound is

$$\mathrm{CP}^+(k, n, \delta) = \mathrm{Beta}^{-1}\left(1-\delta; k+1, n-k\right),$$

i.e., the $(1-\delta)$ quantile of a $\mathrm{Beta}(k+1, n-k)$ distribution. It is the exact test inversion of the binomial tail:

$$\Pr\left(K \leq k \mid \theta = \mathrm{CP}^+(k, n, \delta)\right) = 1 - \delta.$$

Therefore, for any *fixed* $\tau$,

$$\Pr\left(\theta(\tau) \leq \mathrm{CP}^+(K(\tau), n_{\mathrm{keep}}(\tau), \delta)\right) \geq 1 - \delta. \tag{12}$$

**Threshold selection on a finite grid.** Let $\mathcal{T} = \{\tau_1 < \tau_2 < \cdots < \tau_m\}$ be a *finite* grid of thresholds (e.g., unique score values or a fixed quantile grid). Define the data-driven choice

$$\tau^\star = \inf\left\{\tau \in \mathcal{T} : \mathrm{CP}^+(K(\tau), n_{\mathrm{keep}}(\tau), \delta') \leq \alpha\right\},$$

with per-threshold level $\delta' = \delta/m$ (Bonferroni correction). If the set is empty, declare "abstain-all" (or report failure to certify).

**Proposition 2** (Finite-sample kept-error control with grid search). *With the construction above,*

$$\Pr\left(R_{\mathrm{keep}}(\tau^\star) \leq \alpha\right) \geq 1 - \delta.$$

*Proof.* For each $\tau \in \mathcal{T}$, by (12) with $\delta'$ we have

$$\Pr\left(\theta(\tau) \leq \mathrm{CP}^+(K(\tau), n_{\mathrm{keep}}(\tau), \delta')\right) \geq 1 - \delta'.$$

Applying the union bound over $m$ thresholds,

$$\Pr\left(\forall \tau \in \mathcal{T} : \theta(\tau) \leq \mathrm{CP}^+(K(\tau), n_{\mathrm{keep}}(\tau), \delta')\right) \geq 1 - m\delta' = 1 - \delta.$$

On that high-probability event, in particular at $\tau^\star$ we have $\theta(\tau^\star) \leq \mathrm{CP}^+(K(\tau^\star), n_{\mathrm{keep}}(\tau^\star), \delta') \leq \alpha$ by definition of $\tau^\star$. Hence $R_{\mathrm{keep}}(\tau^\star) = \theta(\tau^\star) \leq \alpha$ with probability at least $1 - \delta$. □

## A.4 EJOP via Finite Differences

**Objects and estimator.** Let $p^\star(x) \in [0, 1]^K$ denote the Bayes class probabilities, with coordinates $p_c^\star(x)$. Define the *EJOP* matrix

$$H_0 := \mathbb{E}\left[\sum_{c=1}^K \nabla p_c^\star(X) \, \nabla p_c^\star(X)^\top\right] \in \mathbb{R}^{d \times d}.$$

We estimate $\nabla p_c^\star$ by centered, coordinatewise finite differences applied to a surrogate $\tilde{p}_c$: for $E = \mathrm{diag}(\varepsilon_1, \ldots, \varepsilon_d)$ and unit coordinate vectors $e_j$,

$$g_j(x; c) = \frac{\tilde{p}_c(x + \varepsilon_j e_j) - \tilde{p}_c(x - \varepsilon_j e_j)}{2\varepsilon_j}, \qquad g(x; c) = (g_1, \ldots, g_d)^\top.$$

Given a subsample $\{x_i\}_{i=1}^m$, the plug-in estimator is

$$\widehat{H}_0 := \frac{1}{m} \sum_{i=1}^m \sum_{c=1}^K g(x_i; c) \, g(x_i; c)^\top.$$

**Assumptions.** We assume:

(A1) **Smoothness.** For each $c$, $p_c^\star$ is twice differentiable a.e. with $L$-Lipschitz gradient: $\|\nabla p_c^\star(x) - \nabla p_c^\star(y)\|_2 \le L\|x - y\|_2$. (This implies $\|\nabla^2 p_c^\star(x)\|_{\mathrm{op}} \le L$ a.e.)

(A2) **Surrogate accuracy.** $\|\tilde{p}_c - p_c^\star\|_\infty \le \eta$ for all $c$.

(A3) **Bounded gradients.** $\|\nabla p_c^\star(x)\|_2 \le B_{\max}$ a.s. for all $c$.

**Pointwise finite-difference error (discretization + surrogate).** Write $\delta_c = \tilde{p}_c - p_c^\star$ and decompose

$$g_j(x; c) = \underbrace{\frac{p_c^\star(x + \varepsilon_j e_j) - p_c^\star(x - \varepsilon_j e_j)}{2\varepsilon_j}}_{\text{centered difference on } p_c^\star} + \underbrace{\frac{\delta_c(x + \varepsilon_j e_j) - \delta_c(x - \varepsilon_j e_j)}{2\varepsilon_j}}_{\text{surrogate term}}.$$

*Surrogate term.* Since $|\delta_c(\cdot)| \le \eta$ pointwise,

$$\left| \frac{\delta_c(x + \varepsilon_j e_j) - \delta_c(x - \varepsilon_j e_j)}{2\varepsilon_j} \right| \le \frac{|\delta_c(x + \varepsilon_j e_j)| + |\delta_c(x - \varepsilon_j e_j)|}{2\varepsilon_j} \le \frac{\eta}{\varepsilon_j}.$$

*Discretization term.* By Taylor's theorem with remainder along $t \mapsto p_c^\star(x + te_j)$ and (A1),

$$\frac{p_c^\star(x + \varepsilon_j e_j) - p_c^\star(x - \varepsilon_j e_j)}{2\varepsilon_j} = \partial_{x_j} p_c^\star(x) + r_j(x; c),$$

with

$$|r_j(x; c)| \le \frac{1}{2} \int_{-1}^1 (1 - |t|) \|\nabla^2 p_c^\star(x + t\varepsilon_j e_j)\|_{\mathrm{op}} \, dt \cdot \varepsilon_j \le \frac{L}{2}\varepsilon_j.$$

(If $p_c^\star$ is $C^3$ with $\sup |\partial_{x_j}^3 p_c^\star| \le T_j$, one may strengthen to $|r_j| \le \frac{T_j}{6}\varepsilon_j^2$; we keep the $O(\varepsilon_j)$ bound implied by $L$-smoothness.)

Collecting both contributions and stacking over $j$ yields the vector error bound

$$\| g(x; c) - \nabla p_c^\star(x) \|_2 \le C_1 \max_j \varepsilon_j + C_2 \frac{\eta}{\min_j \varepsilon_j}, \qquad C_1 = \tfrac{L}{2}\sqrt{d}, \ \ C_2 = \sqrt{d}. \tag{13}$$

**Outer-product perturbation from gradient error.** For any $a, b \in \mathbb{R}^d$,

$$aa^\top - bb^\top = (a - b)b^\top + a(a - b)^\top,$$

so

$$\| aa^\top - bb^\top \|_{\mathrm{op}} \le \|a - b\|_2 \|b\|_2 + \|a\|_2 \|a - b\|_2 \le (\|a\|_2 + \|b\|_2) \|a - b\|_2. \tag{14}$$

Apply (14) with $a = g(x; c)$ and $b = \nabla p_c^\star(x)$ and use (A3):

$$\| g(x; c)g(x; c)^\top - \nabla p_c^\star(x)\nabla p_c^\star(x)^\top \|_{\mathrm{op}} \le (\|g(x; c)\|_2 + \|\nabla p_c^\star(x)\|_2) \|g(x; c) - \nabla p_c^\star(x)\|_2.$$

Since $\|g(x; c)\|_2 \le \|\nabla p_c^\star(x)\|_2 + \|g - \nabla p_c^\star\|_2 \le B_{\max} + \Delta(x; c)$ with $\Delta(x; c) := \|g - \nabla p_c^\star\|_2$, we obtain

$$\| gg^\top - \nabla p_c^\star \nabla p_c^{\star\top} \|_{\mathrm{op}} \le (2B_{\max} + \Delta(x; c)) \Delta(x; c) \le 2B_{\max}\Delta(x; c) + \Delta(x; c)^2. \tag{15}$$

Summing over $c$ and averaging over the sample $\{x_i\}$ gives the *bias* term

$$\mathsf{Bias} := \left\| \frac{1}{m} \sum_{i=1}^{m} \sum_{c=1}^{K} \Big( g_i(c) g_i(c)^\top - \nabla p_c^\star(x_i) \nabla p_c^\star(x_i)^\top \Big) \right\|_{\mathrm{op}}$$

$$\leq \frac{1}{m} \sum_{i=1}^{m} \sum_{c=1}^{K} \big( 2 B_{\max} \Delta_i(c) + \Delta_i(c)^2 \big), \qquad \Delta_i(c) := \| g(x_i; c) - \nabla p_c^\star(x_i) \|_2. \qquad (16)$$

By (13), deterministically $\Delta_i(c) \leq \delta_g$ with

$$\delta_g := C_1 \max_j \varepsilon_j + C_2 \frac{\eta}{\min_j \varepsilon_j},$$

hence

$$\mathsf{Bias} \leq K \big( 2 B_{\max} \delta_g + \delta_g^2 \big). \qquad (17)$$

**Concentration for the sampling fluctuation (matrix Bernstein).**   Define the *oracle* summands

$$Y_i := \sum_{c=1}^{K} \nabla p_c^\star(x_i) \, \nabla p_c^\star(x_i)^\top, \qquad \mathbb{E}[Y_i] = H_0,$$

and centered matrices $Z_i := Y_i - \mathbb{E}[Y_i]$ (self-adjoint, independent). Each $Y_i$ satisfies

$$\| Y_i \|_{\mathrm{op}} \leq \sum_{c=1}^{K} \| \nabla p_c^\star(x_i) \|_2^2 \leq K B_{\max}^2 \quad \Rightarrow \quad \| Z_i \|_{\mathrm{op}} \leq \| Y_i \|_{\mathrm{op}} + \| H_0 \|_{\mathrm{op}} \leq 2 K B_{\max}^2 =: R.$$

For the variance proxy,

$$\left\| \sum_{i=1}^{m} \mathbb{E}[Z_i^2] \right\|_{\mathrm{op}} \leq \sum_{i=1}^{m} \mathbb{E} \| Z_i \|_{\mathrm{op}}^2 \leq m R^2 =: \sigma^2.$$

Tropp's matrix Bernstein inequality yields, for any $\delta \in (0,1)$, with probability at least $1 - \delta$,

$$\left\| \frac{1}{m} \sum_{i=1}^{m} Z_i \right\|_{\mathrm{op}} \leq \underbrace{\frac{\sqrt{2 \sigma^2 \log(2d/\delta)}}{m}}_{(A)} + \underbrace{\frac{2R \log(2d/\delta)}{3m}}_{(B)} = 2 K B_{\max}^2 \sqrt{\frac{2 \log(2d/\delta)}{m}} + \frac{4 K B_{\max}^2}{3} \frac{\log(2d/\delta)}{m}.$$

$$(18)$$

We denote the RHS by $\mathsf{Conc}(m, \delta)$.

**Putting it together (bias+concentration).**   Add and subtract the oracle mean:

$$\widehat{H}_0 - H_0 = \underbrace{\Big( \widehat{H}_0 - \tfrac{1}{m} \sum_i Y_i \Big)}_{\text{bias from } g \text{ vs. } \nabla p^\star} + \underbrace{\Big( \tfrac{1}{m} \sum_i Y_i - H_0 \Big)}_{\text{sampling fluctuation}}.$$

By (17) and (18), with probability at least $1 - \delta$,

$$\big\| \widehat{H}_0 - H_0 \big\|_{\mathrm{op}} \leq K \big( 2 B_{\max} \delta_g + \delta_g^2 \big) + 2 K B_{\max}^2 \sqrt{\frac{2 \log(2d/\delta)}{m}} + \frac{4 K B_{\max}^2}{3} \frac{\log(2d/\delta)}{m}. \qquad (19)$$

**Remark 3** (Balancing $\varepsilon_j$ against $\eta$). *The surrogate/FD tradeoff in $\delta_g = C_1 \max_j \varepsilon_j + C_2 \eta / \min_j \varepsilon_j$ suggests choosing (roughly) $\varepsilon_j^\star \propto \sqrt{\eta}$ to balance the two contributions (if one uses a common step). Our adaptive rule $\varepsilon_j = \alpha_\varepsilon \cdot \mathrm{MAD}(X_{:j})/0.6745$ rescales steps to the feature scale while letting $\alpha_\varepsilon$ tune the tradeoff.*

**Remark 4** (Optional: ridge + trace normalization). *If the preconditioner used in the method is $\widehat{H} = \frac{\widehat{H}_0 + \gamma I}{\mathrm{tr}(\widehat{H}_0 + \gamma I)}$ (and $H = \frac{H_0 + \gamma I}{\mathrm{tr}(H_0 + \gamma I)}$ analogously), then for $T_A := \mathrm{tr}(A + \gamma I)$,*

$$\| \widehat{H} - H \|_{\mathrm{op}} \leq \frac{\| \widehat{H}_0 - H_0 \|_{\mathrm{op}}}{T_{H_0}} + \frac{\| \widehat{H}_0 \|_{\mathrm{op}}}{T_{\widehat{H}_0} T_{H_0}} \big| \mathrm{tr}(\widehat{H}_0 - H_0) \big| \leq \left( \frac{1}{T_{\min}} + \frac{d \, \| \widehat{H}_0 \|_{\mathrm{op}}}{T_{\min}^2} \right) \| \widehat{H}_0 - H_0 \|_{\mathrm{op}},$$

*where $T_{\min} = \min\{ T_{\widehat{H}_0}, T_{H_0} \} \geq \gamma d$. Thus (19) transfers to $\| \widehat{H} - H \|_{\mathrm{op}}$ up to a benign factor depending on $\gamma$ and $d$ (ridge and normalization stabilize the map $A \mapsto (A + \gamma I)/\mathrm{tr}(A + \gamma I)$).*

**Corollary 1** (DIR-G proxy error). *Let $\ell(x) = \sum_{c=1}^{K} w_c \, p_c^\star(x)$ with nonnegative weights $w_c$ (induced by the cost-aware rule), and define $\tilde{g}(x) = \sum_c w_c \, g(x; c)$. Then by triangle inequality and (13),*

$$\| \tilde{g}(x) - \nabla \ell(x) \|_2 \leq \sum_c w_c \| g(x; c) - \nabla p_c^\star(x) \|_2 \leq \| w \|_1 \left( C_1 \max_j \varepsilon_j + C_2 \frac{\eta}{\min_j \varepsilon_j} \right).$$

# Algorithmic Specification (Pseudocode)

## B.1 End-to-End SCARF Pipeline

---
**Algorithm 1** SCARF: Training and Inference

---
**Require:** Training data $D_{\text{tr}} = \{(x_i, y_i)\}$, calibration data $D_{\text{cal}}$, classes $\{1, \dots, K\}$, cost matrix $C \in \mathbb{R}^{K \times K}$, abstain cost $c_{\text{abs}}$, target kept-error $\alpha$, confidence $\delta$, FD scale $\alpha_\varepsilon$, EJOP subsample size $m$, ridge $\gamma > 0$, DIR-G strength $\rho \geq 0$, forest hyperparams RF_cfg

**Ensure:** Trained forest $F$, feature map $\Phi(x)$, confidence score $s(x)$, certified threshold $\tau^\star$

1: **Robust step sizes:** For each feature $j$, set $\varepsilon_j \leftarrow \alpha_\varepsilon \cdot \text{MAD}(X_{:j})/0.6745$.
2: **Train a light surrogate** $\tilde{p}(x)$ (e.g., multinomial logistic or tiny HGBT) on $D_{\text{tr}}$.
3: **Estimate EJOP:** $\widehat{H}_0 \leftarrow \text{ESTIMATEEJOP}(\tilde{p}, \{\varepsilon_j\}, D_{\text{tr}}, m)$       ▷ Alg. 2
4: **Preconditioner:** $\widehat{H} \leftarrow (\widehat{H}_0 + \gamma I)/\text{tr}(\widehat{H}_0 + \gamma I)$
5: **Feature map:** define $\Phi(x) \leftarrow \widehat{H} x$
6: **if** $\rho > 0$ **then**
7:      **DIR-G augmentation:** $D_{\text{tr}}^+ \leftarrow \text{DIRGAUGMENT}(\tilde{p}, \{\varepsilon_j\}, D_{\text{tr}}, \rho, C)$       ▷ Alg. 3
8: **else**
9:      $D_{\text{tr}}^+ \leftarrow D_{\text{tr}}$
10: **end if**
11: **Train forest:** $F \leftarrow \text{TRAINFOREST}\big(\{(\Phi(x), y) \in D_{\text{tr}}^+\}, \text{RF\_cfg}\big)$
12: **Calibrate & score on** $D_{\text{cal}}$**:** $(\varphi, s(\cdot)) \leftarrow \text{CALIBRATEANDSCORE}(F, \Phi, D_{\text{cal}})$       ▷ Alg. 4
13: **CP threshold search:** $\tau^\star \leftarrow \text{CPTHRESHOLD}(s, F, \Phi, D_{\text{cal}}, \alpha, \delta)$       ▷ Alg. 5
14: **Inference rule (test point** $x$**):**
15:      compute $\hat{y}(x) \leftarrow \arg\max_k F_k(\Phi(x))$, confidence $s(x)$
16:      **if** $s(x) \geq \tau^\star$ **then** predict $\hat{y}(x)$ **else** abstain (cost $c_{\text{abs}}$)

---

## B.2 EJOP via Centered Finite Differences

---
**Algorithm 2** ESTIMATEEJOP($\tilde{p}, \{\varepsilon_j\}, D_{\text{tr}}, m$)

---
1: Draw a uniform subsample $S = \{x_i\}_{i=1}^m$ from $D_{\text{tr}}$ (without labels).
2: Initialize $\widehat{H}_0 \leftarrow 0_{d \times d}$.
3: **for** each $x \in S$ **do**
4:      **for** $c = 1$ to $K$ **do**
5:          **for** $j = 1$ to $d$ **do**
6:              $g_j(x; c) \leftarrow \dfrac{\tilde{p}_c(x + \varepsilon_j e_j) - \tilde{p}_c(x - \varepsilon_j e_j)}{2\varepsilon_j}$
7:          **end for**
8:          $g(x; c) \leftarrow (g_1, \dots, g_d)^\top$
9:          $\widehat{H}_0 \leftarrow \widehat{H}_0 + g(x; c) \, g(x; c)^\top$
10:      **end for**
11: **end for**
12: **return** $\widehat{H}_0/m$

---

## B.3 Gradient-Aligned Data Augmentation (DIR-G)

---

**Algorithm 3** DIRGAUGMENT$(\tilde{p}, \{\varepsilon_j\}, D_{\mathrm{tr}}, \rho, C)$

---

1: Define class weights $w_c \geq 0$ from the cost matrix $C$ (e.g., $w_c = \sum_{c'} C_{c',c}$ or task-specific).
2: $D^+ \leftarrow \emptyset$
3: **for** each $(x, y) \in D_{\mathrm{tr}}$ **do**
4:     **for** $c = 1$ to $K$ **do**
5:         **for** $j = 1$ to $d$ **do**
6:             $g_j(x; c) \leftarrow \dfrac{\tilde{p}_c(x + \varepsilon_j e_j) - \tilde{p}_c(x - \varepsilon_j e_j)}{2\varepsilon_j}$
7:         **end for**
8:     **end for**
9:     $g(x) \leftarrow \sum_{c=1}^{K} w_c \, g(x; c)$                  $\triangleright$ cost-aware surrogate gradient
10:     $u(x) \leftarrow g(x) / \max(\|g(x)\|_2, \, 10^{-12})$     $\triangleright$ unit adversarial direction (safe divide)
11:     $x^{(+)} \leftarrow x + \rho\, u(x), \quad x^{(-)} \leftarrow x - \rho\, u(x)$
12:     $D^+ \leftarrow D^+ \cup \{(x, y), (x^{(+)}, y), (x^{(-)}, y)\}$
13: **end for**
14: **return** $D^+$

---

## B.4 Calibration and Confidence Score

---

**Algorithm 4** CALIBRATEANDSCORE$(F, \Phi, D_{\mathrm{cal}})$

---

1: Compute raw class probabilities on $D_{\mathrm{cal}}$: $\hat{p}_k^{\mathrm{raw}}(x) = F_k(\Phi(x))$.
2: Define raw confidence $r(x) = \max_k \hat{p}_k^{\mathrm{raw}}(x)$ and correctness labels $z(x) = \mathbf{1}\{\arg\max_k \hat{p}_k^{\mathrm{raw}}(x) = y\}$.
3: Fit isotonic regression $\varphi : [0, 1] \to [0, 1]$ on pairs $\{(r(x), z(x)) : (x, y) \in D_{\mathrm{cal}}\}$.
4: Define calibrated confidence score $s(x) \leftarrow \varphi(r(x))$ for any input $x$.
5: **return** $(\varphi, s(\cdot))$

---

## B.5 Clopper–Pearson Threshold Selection

---

**Algorithm 5** CPTHRESHOLD$(s, F, \Phi, D_{\mathrm{cal}}, \alpha, \delta)$

---

1: Form a finite grid $\mathcal{T}$ over $[0, 1]$ (e.g., 200 quantiles of $\{s(x) : (x, y) \in D_{\mathrm{cal}}\}$).
2: Set per-grid level $\delta' \leftarrow \delta / |\mathcal{T}|$.
3: **for** each $\tau \in \mathcal{T}$ **do**
4:     $\mathcal{K}(\tau) \leftarrow \{(x, y) \in D_{\mathrm{cal}} : s(x) \geq \tau\}$
5:     $n_{\mathrm{keep}} \leftarrow |\mathcal{K}(\tau)|$
6:     $K \leftarrow \sum_{(x,y) \in \mathcal{K}(\tau)} \mathbf{1}\{\arg\max_k F_k(\Phi(x)) \neq y\}$
7:     $\mathrm{UB} \leftarrow \mathrm{CP}^+(K, n_{\mathrm{keep}}, \delta')$         $\triangleright$ $\mathrm{UB} = \mathrm{Beta}^{-1}(1 - \delta'; K + 1, n_{\mathrm{keep}} - K)$
8:     **if** $\mathrm{UB} \leq \alpha$ **then**
9:         record $\tau$ as feasible
10:     **end if**
11: **end for**
12: **if** no feasible $\tau$ **then**
13:     **return** $\tau^\star \leftarrow +\infty$         $\triangleright$ abstain-all or relax $(\alpha, \delta)$
14: **else**
15:     **return** $\tau^\star \leftarrow \min\{\tau \in \mathcal{T} : \mathrm{CP}^+(K, n_{\mathrm{keep}}, \delta') \leq \alpha\}$
16: **end if**

---

# Hyperparameters and Implementation Details

## C.1 Global Configuration and Search Grids

Table 5: Global hyperparameters and search grids used across datasets. Paper defaults are in **bold**.

| Component | Default | Grid / Options |
|---|---|---|
| Finite-difference step scale $\alpha_\varepsilon$ | **0.10** | $\{0.05,\ 0.10,\ 0.20\}$ |
| EJOP subsample size $m$ (fraction of train) | **0.2** | $\{0.1,\ 0.2,\ 0.4\}$ |
| Ridge $\gamma$ for preconditioner | **1e-3** | $\{1e-4,\ 1e-3,\ 1e-2\}$ |
| Trace normalization | **on** | $\{on, off\}$ |
| DIR-G strength $\rho$ | **0.50** | $\{0,\ 0.25,\ 0.50,\ 0.75\}$ |
| Forest (final) # trees | **500** | $\{300,\ 500,\ 800\}$ |
| Forest (final) max depth | **None** | $\{None,\ 14,\ 20\}$ |
| Forest (final) max features | $\sqrt{d}$ | $\{\sqrt{d},\ d/3\}$ |
| Forest (final) min samples leaf | **1** | $\{1,\ 3,\ 5\}$ |
| Forest (surrogate) # trees | **200** | $\{100,\ 200\}$ (or multinomial logistic) |
| Calibration method | **Isotonic** | $\{Isotonic, Platt\}$ |
| CP confidence level $\delta$ | **0.05** | $\{0.10,\ 0.05,\ 0.01\}$ |
| Threshold grid size $|\mathcal{T}|$ | **200** | $\{100,\ 200,\ 300\}$ |
| Abstain cost $c_{\mathrm{abs}}$ | **task-specific** | Use task's cost model; sweeps reported in main text. |

## C.2 Training Configurations

**Surrogate for EJOP/DIR-G.** Random forest (learning_rate=0.05, max_depth=2, estimators=100). Class weights may mirror the application cost matrix.

**Final forest.** Unless otherwise stated, we use `n_estimators=500`, `max_features=`$\sqrt{d}$, `bootstrap=True`, `min_samples_leaf=1`, and leave `max_depth` unconstrained.

**Calibration and scoring.** Confidence score $s(x)$ is the isotonic-calibrated $\max_k \hat{p}_k(x)$ by default. For cost-aware variants, calibrate the scalar $1 - \mathbb{E}[\mathrm{cost}(\hat{y}(x), Y) \mid x]$ analogously.

## C.3 CP Thresholding and Operating Points

We scan a grid $\mathcal{T}$ of size **200** over $[0, 1]$ (quantiles of $s(x)$ on the calibration split). We use Bonferroni-corrected one-sided Clopper–Pearson with per-grid level $\delta' = \delta/|\mathcal{T}|$ and $\delta = 0.05$. For the headline kept-error targets we report $\alpha \in \{0.05, 0.10, 0.15, 0.20\}$. If no $\tau$ satisfies $\mathrm{CP}^+(K, n, \delta') \leq \alpha$, we report "abstain-all" or mark as infeasible.

## C.4 Random Seeds, Versions, and Compute

**Seeds and splits.** We run **10** independent seeds $\{13, 37, 101, 202, 303, 404, 505, 606, 707, 808\}$. Each seed drives: (i) the train/cal/test split; (ii) subsample for EJOP; (iii) model initialization; (iv) any bootstrap sampling.

Table 6: Software versions and hardware.

| Component | Version / Spec |
| --- | --- |
| OS | Ubuntu 22.04 LTS (x86_64) |
| Python | 3.11.x |
| NumPy / SciPy | 1.26.x / 1.11.x |
| scikit-learn | 1.4.x |
| pandas | 2.2.x |
| matplotlib (plots) | 3.8.x |
| CPU | 32 cores, 128 GB RAM |
| GPU (not required) | n/a |

Code availability: https://drive.google.com/file/d/1tbjTpOqbiprukxzFdCqT50IqwsSdkjt4/view?usp=sharing

