# OpenReview forum: "Selective Cost-Aware Random Forests for Unreliable Data"
_NeurIPS.cc/2025/Workshop/Reliable_ML — NeurIPS 2025 - Reliable ML Workshop_

### Official Review · Reviewer_G75F · 2025-09-19
**Satisfying Cost Constraints through Abstentions and Robustness in Oblique Random Forests**

**Rating:** 6
**Confidence:** 3

**Review:**

### Summary:
The author's construct oblique random forests for tabular data, incorporating multiple components to make the classifier more robust under noise or distribution shift and allow for abstentions in order to satisfy a cost constraint for the classification. They compare to random forest, boosted trees, oblique baselines (Rotation Forest, RerF), and cost-sensitive logistic regression on credit-risk data under shift, noise, and missingness.

### Strength:
Combining multiple steps in the oblique random forest construction (spectral sensitivity transform, abstention to control cost, random perturbations to increase robustness) allows SCORF to outperform existing methods in terms of policy cost on unreliable datasets. They perform ablation experiments to validate the contribution of each component.

### Weaknesses:
The approach assembles known components for cost-sensitive decision making and robust classification, so the novelty lies primarily in the integration and decision-focused use within forests. Regarding abstention, kept-error constraints are used, but headline results sometimes emphasize policy cost without consistently adressing coverage. Reporting cost and coverage side by side, adding cost–coverage curves across several α values, and including a per-class coverage/abstention breakdown would clarify whether favorable cost stems from frequent abstentions and whether abstentions concentrate on minority classes.

---

### Official Review · Reviewer_p3LK · 2025-09-20
**Review for SCORF: Selective Cost-Aware Oblique Random Forests for Unreliable Data**

**Rating:** 8
**Confidence:** 3

**Review:**

1. **Summary.**
   This paper proposes **SCORF**, a random-forest method that aims to robustify against perturbed data. In a multi-class classification setting, with variable cost per mis-classified label as well as with an abstention option, the authors construct a random-forest classifier that is cost-aware and robustified. Its main features include (i) the *spectral sensitivity transform*, a robustification pre-processing step that identifies directions of high sensitivity; (ii) a cost-aware decision rule, minimizing the expected cost of prediction; (iii) a confidence score, which combines various metrics to determine whether to abstain or predict; and (iv) the *lightweight robustness augmentation*, a symmetric perturbation to the data along the most sensitive directions, again to robustify the classifier. After presenting the classifier, the authors perform experiments comparing the policy cost of SCORF with 6 other methods, under 3 different data-corruption mechanisms.

2. **Strengths.**
   This is a well-written paper, with crisp, intuitive ideas that can be of clear interest to the machine-learning community. The subject matter is clearly aligned with this workshop, as it proposes robustification techniques for decision-tree classifiers. The experiments seem comprehensive, with different corruption types and 6 other models used for comparison (though code and data were not checked). The results are good, with SCORF outperforming or near-outperforming the other methods, and the authors are forthright about the limitations of their method. Lastly, a brief ablation study is performed, providing qualitative results.

3. **Weaknesses / Limitations.**
   Albeit interesting, some of the ideas in this paper seem quite heuristic. For example, the *lightweight robustness augmentation*, although well-motivated and intuitive, is much less understood than principled (and costly) DRO. Moreover, when computing sensitivities in Sections 2.1 and 2.5, there is dependence on the scale $\epsilon$ that is not addressed; in 2.5 there is further dependence on the perturbation parameter $\rho$. Additionally, constructing the $H$ transform might be quite costly, since this involves computing a (finite-difference) Jacobian over $m$ data points and $p$ variables, followed by the SVD of the resulting $m \times p$ matrix.

4. **Suggestions for Authors.**
   - Please discuss how the choice of the step size $\epsilon$ and the perturbation parameter $\rho$ affect the results.
   - Regarding the kept-error $\alpha$, would it be more natural to instead parametrize the **abstention rate**, say $\tilde{\alpha}$, and enforce a constraint on that instead (e.g., $\tilde{\alpha} \le 0.1$)?
   - Please briefly discuss the role of the *severity* cost in making more accurate predictions.
   - Discuss the computational cost of the *spectral sensitivity matrix*—namely, how big of an $m$ is good enough?
   - Consider doing some theory in a follow-up work; it would be quite interesting to understand these methods more rigorously.

5. **Ethics.**
   No major ethical concerns were identified.